# The Anticancer Effects of Flavonoids through miRNAs Modulations in Triple-Negative Breast Cancer

**DOI:** 10.3390/nu13041212

**Published:** 2021-04-07

**Authors:** Getinet M. Adinew, Equar Taka, Patricia Mendonca, Samia S. Messeha, Karam F. A. Soliman

**Affiliations:** Division of Pharmaceutical Sciences, College of Pharmacy and Pharmaceutical Sciences, Institute of Public Health, Florida A&M University, Tallahassee, FL 32307, USA; getinet1.mequanint@famu.edu (G.M.A.); equar.taka@famu.edu (E.T.); patricia.mendonca@famu.edu (P.M.); samia.messeha@famu.edu (S.S.M.)

**Keywords:** cancer, flavonoid, microRNA, triple-negative breast cancer

## Abstract

Triple- negative breast cancer (TNBC) incidence rate has regularly risen over the last decades and is expected to increase in the future. Finding novel treatment options with minimum or no toxicity is of great importance in treating or preventing TNBC. Flavonoids are new attractive molecules that might fulfill this promising therapeutic option. Flavonoids have shown many biological activities, including antioxidant, anti-inflammatory, and anticancer effects. In addition to their anticancer effects by arresting the cell cycle, inducing apoptosis, and suppressing cancer cell proliferation, flavonoids can modulate non-coding microRNAs (miRNAs) function. Several preclinical and epidemiological studies indicate the possible therapeutic potential of these compounds. Flavonoids display a unique ability to change miRNAs’ levels via different mechanisms, either by suppressing oncogenic miRNAs or activating oncosuppressor miRNAs or affecting transcriptional, epigenetic miRNA processing in TNBC. Flavonoids are not only involved in the regulation of miRNA-mediated cancer initiation, growth, proliferation, differentiation, invasion, metastasis, and epithelial-to-mesenchymal transition (EMT), but also control miRNAs-mediated biological processes that significantly impact TNBC, such as cell cycle, immune system, mitochondrial dysregulation, modulating signaling pathways, inflammation, and angiogenesis. In this review, we highlighted the role of miRNAs in TNBC cancer progression and the effect of flavonoids on miRNA regulation, emphasizing their anticipated role in the prevention and treatment of TNBC.

## 1. Introduction

Globally, breast cancer (BC) is the major and most common repeatedly diagnosed cancer in women, which accounts for 30% of new female cancer cases [1], and also the second cause of death in women worldwide [2]. Approximately 1 million breast cancer cases are diagnosed annually worldwide [3]. In the United States, more than 276,000 new breast cancer cases were estimated by the end of 2020, and 12.9% of all women will be diagnosed with breast cancer over their lifetime [4,5,6,7]. Approximately 15% of breast cancers are categorized as triple- negative breast cancer (TNBC), characterized by a poor prognosis, early relapse, distant recurrence, unresponsiveness to conventional treatment, aggressive tumor growth, aggressive clinical demonstration, and lowest survival rate [8]. Compared with other BC subtypes, TNBC is more often associated with hereditary conditions. Evidence showed that among newly diagnosed BC patients, around 35% of BC suppressor protein1 (BRCA1) and 8% of BC suppressor protein2 (BRCA2) mutations in this population were TNBC [9]. Lack of progesterone (PR), estrogen (ER), and human epidermal growth factor receptor 2 (HER2) receptors are the major features of TNBC [10]. Recently, according to intrinsic gene signature, TNBC can be classified into six main types: basal-like 1 and 2, mesenchymal stem-like, immunomodulatory, mesenchymal, and luminal androgen receptor [11]. Of the TNBC cases, an estimated 75% are basal-like [12]. The prevalence of TNBC in African American women is higher than non-African American women. Indeed, 39% of African American premenopausal women diagnosed with BC are TNBC [13]. Previously reported studies revealed the continuous increase in BC incidence rate over the last decades and in the future [14].

Chemotherapy and radiotherapy are the two most common treatment strategies for TNBC patients in the early or advanced stages [15]. Compared to hormone receptor-positive patients, TNBC patients initially respond to conventional chemotherapy. However, the frequent disease relapse results in the worst outcome and low survival rate due to high metastasis rates and lack of effective treatment after relapse [16,17]. Although chemoresistance is a challenge that accounts for a significant share of drug failures [18], chemotherapy remains the primary cancer treatment approach. It is the only agent approved by the Food and Drug Administration (FDA) in treating nonmetastatic TNBC [19]. Even though the mechanism of resistance depends on the chemotherapeutic agent and patient; drug inactivation, drug target alteration, DNA damage repair, cell death inhibition, cancer cell heterogeneity, epigenetic alteration, and epithelial–mesenchymal transition or combination of these are the major direct or indirect contributing factor for developing resistance against cancer chemotherapeutic agents [18]. In TNBC cells, epigenetic mechanisms are implicated in chemotherapy resistance. For instance, an inherent defect in drug uptake and a lack of reduced folate carrier expression is the main cause of methotrexate resistance in MDA-MB-231 cells. However, treating MDA-MB-231 cells with DNA methylation inhibitor or reduced folate carrier cDNA was previously reported to restore methotrexate uptake and enhance sensitivity to methotrexate [20].

MicroRNAs were identified to be correlated with chemoresistance in TNBC. For instance, resistance to neoadjuvant chemotherapy was strongly linked to upregulated miR-181a [21]. Similarly, in the MDA-MB-231 cell line, upregulation of miR-21-3p, miR-155-5p, miR-181a-5p, miR-181b-5p, 183-5p and downregulation of miR-10b-5p, miR-31-5p, miR-125b-5p, miR-195-5p, and miR-451a were associated with doxorubicin resistance [22,23]. Moreover, downregulation of miR-200c was associated with doxorubicin resistance, poor response to radiotherapy, and increased multidrug resistance mediated gene expression [24]. Taken all together, chemoresistance is still a challenge in preventing and treating TNBC, and finding the best options is needed to manage the disease by developing drugs that combat the resistance gene or any target molecules of TNBC, miRNAs.

This review focuses on the anticancer properties of flavonoids in TNBC through miRNA regulation, utilizing compounds that target various pathways involved in cancer initiation, growth, proliferation, differentiation, survival, migration, invasiveness metastasis, and epithelial-to-mesenchymal transition (EMT). Additionally, the miRNA mechanism of action on cancer proliferation, cell cycle, immune system, mitochondrial dysregulation, modulating signaling pathways, inflammation, angiogenesis, invasion and metastasis, and apoptosis will be examined.

## 2. The Microenvironment of TNBC

The tumor microenvironment (TME) refers to the cellular environment in which tumor cells exist. The interaction between cancer and non-cancerous cells is a critical regulator of carcinogenesis that controls the sequence of cancer cell growth and progression. Indeed, TME highly determines the initiation, development, proliferation, angiogenesis, invasiveness, metastasis, and tumor behavior progression. These biological properties nominated TME as a very promising target for treating cancer cells, including TNBC [25]. It encompasses the surrounding immune cells, extracellular matrix, blood vessels, lymphocytes, fibroblasts, adipocytes, inflammatory cells, and signaling molecules [26]. MicroRNAs are involved in regulating various signaling pathways within the tumor microenvironment [27]. 

Among the cellular components in TME, endothelial cells play a crucial role in tumor development, protect tumor cells from the immune system, and activate new angiogenic vessels that offer nutritional support for growth and development [28]. The expression of vascular endothelial growth factor (VEGF), a pro-angiogenesis factor that binds to the endothelial cell surface receptor, is significantly higher in TNBC than in non-TNBC tumors [29]. Evidence showed that VEG-induced adhesion and migration of MDA-MB-231 cells occurs in co-transfected with endothelial cells [30]. Endothelial-specific micro RNAs like miR-126 are reported to promote angiogenesis by upregulating VEGF [31]. Fibroblast is another component of TME that allows tumor cell migration from the original location into the bloodstream for systemic metastasis and is involved in passing endothelial cells undergoing angiogenesis in the tumor [32]. Cancer-associated fibroblasts (CAF) are correlated with increased tumor progression by enhancing activation of TGF-β in the MDA-MB-231 TNBC cell line [33]. Moreover, coculturing basal-like cancer cells with fibroblasts induced the expression of various interleukins and chemokines such as IL-6, IL-8, CXCL1, and CXCL3 [34] and enhance metastasis through upregulating matrix metalloproteinase-9 (MMP-9) [35]. Additionally, α-smooth muscle actin (α-SMA), a specific biomarker used to detect CAFs in the tumor, increases in MDA-MB-231 cells after co-culturing with CAF compared to positive breast cancer cells. In parallel, genes that encode CAFs [36] and genes expressed by CAFs [37] are associated with chemoresistance. A study conducted on breast tumor tissues identified 11 dysregulated miRNAs in CAFs, in which miR-31-3p, miR-221-5p, and miR-221-3p were upregulated let-7g, miR-26b, miR-10, miR-141, miR-200b, miR-200c, miR-205, and miR-342 were downregulated [38].

The immune system cells, such as macrophages, lymphocytes, and granulocytes, are strongly involved in various immune responses and inflammatory reactions that enhance tumor cell survival [39]. Tumor-associated macrophages enhance tumor progression by stimulating tumor cell migration, angiogenesis, and extravasation at the metastatic area and reduce antitumor immunity. Several studies report dysregulation of miRNA in tumor-associated macrophages including, miR-27a, miR-29-b-1, miR-132, miR-193b, and miR-222 [40,41]. In general, all these data suggested that miRNAs can regulate TME components that promote tumor progression in TNBC cases.

## 3. Epigenetic Modification and TNBC

Myriads of studies have already reported that epigenetic alteration is a prominent feature in cancer initiation and development. Cancers, including TNBC, are driven by the accumulation of genetic abnormalities involving mutations in tumor suppressors and/or oncogenes [42]. However, cancer initiation and progression are a multistep process involving epigenetic and genetic changes [43]. Various epigenetic modifications with diagnostic, prognostic, or therapeutic significance have previously been described in many malignancies, including TNBC [44].

Several epigenetic modifications have been reported in TNBC, including histone modification (e.g., acetylation, methylation, phosphorylation, sumoylation, ubiquitylation), DNA methylation, and noncoding RNAs [45], as well as chromatin remodeling, nucleosome positioning, and chromosomal looping are the common epigenetic alterations. However, DNA methylation, histone modification, and miRNA are the major epigenetic modifications that interfere with genes’ expression and are thus strongly associated with cancer development [46]. Recently, approaching epigenetic modifications in treating TNBC have increased attention. DNA methyltransferase inhibitors and Histone deacetylase inhibitors were shown to enhance antitumor activity in TNBC via epigenetic alterations of EMT [47].

It is believed that miRNAs are susceptible to epigenetic modulation [48,49]. MicroRNAs are important in silencing the expression of genes in TNBC either by sup-pressing or activating various genes at the pre- and post-transcriptional levels, respectively. Even though miRNAs are observed in exons of genes, most are located in endo-nuclear noncoding regions, such as introns of protein-coding genes [50]. Since most human miRNAs are encoded in fragile chromosomal regions sensitive to deletion, amplification, or translocation during initiation and development of cancer, any epigenetic and genetic alterations lead to deregulation of miRNA [51]. Changes can vary microRNAs function and transcription in the epigenetic modifications, hereditary mutations in the DNA encoding the miRNA, activities of proteins involved in the process of miRNAs biogenesis [52], single nucleotide polymorphisms in the miRNA sequence [53], or mutations in miRNA genes [54]. In general, the provided evidence indicates that miRNAs are significantly involved in the process of TNBC carcinogenesis.

## 4. MicroRNA Biogenesis and Function

MicroRNAs (miRNAs) are non-coding, small endogenous, single-chain RNAs composed of 21–23 nucleotides, expressed in most organisms [55], and are important to regulate gene expression post-translationally [56]. MicroRNAs account for 1–5% of the total human genome (approximately 28,000 miRNAs), regulating at least 30% of protein-coding genes [57]. The miRNA biogenesis process is initiated by transcribing primary miRNAs (pri-miRNAs) through RNA polymerase II/III in the nucleus. Post-transcriptional modifications such as capped with 7-methyl guanosine and polyadenylated are taken place on pri-miRNA to form precursor microRNA (pre-miRNAs). There-after, pre-miRNAs move into the cytoplasm via exportin 5, with stem-loop structures, and are processed into mature miRNAs by Drosha, Dicer (RNase III), RNA polymerase III, and other related molecules [58]. These mature miRNAs then bind the RNA-induced silencing complex (RISC), and the resulting co-complex directly binds the 3′-untranslated regions (3′-UTRs) of target mRNAs, acting as a suppressor of translation or helping to increase degradation [59]. Thus, dependent upon the target mRNAs’ identity, miRNAs control various physiological functions, including cell growth, proliferation, differentiation, development, reproduction, apoptosis, metabolism, and its abnormal expression in various cancers, including TNBC [60].

## 5. Types of miRNA in TNBC

Depending on their target’s cell function, miRNAs could serve as either a tumor suppressor or oncogene [61]. Upregulation of specific mRNAs may result in the repression of tumor suppressor gene expression. Simultaneously, down-regulation could lead to an increased oncogene expression, which both induce malignant effects on cell differentiation, proliferation, and apoptosis, resulting in cancer cell growth and progression. A single mRNA can target multiple miRNAs, and similarly, one miRNA can have several mRNA targets. The oncogenic miRNAs could be mediated by targeting many tumor suppressor molecules. Moreover, tumor suppressor miRNAs may inhibit tumor cell initiation, proliferation, metastasis, or induction of apoptosis by suppressing several signaling molecules. MicroRNAs involved in TNBC carcinogenesis processes which were extensively described in the literature are summarized below.

### 5.1. Oncogenic miRNAs

Numerous studies showed that a plethora of miRNAs have shown oncogenic activity in many cancers. For instance, miR-155 is the most frequent upregulated type of microRNA in BC. MiR-155 is involved in tumor cell survival, growth, invasion, migration, EMT, and immune response. In BC, overexpression of miR-155 significantly inhibited the tumor suppressor gene suppressor of cytokine signaling 1 (socs1) and leads to constitutive activation of signal transducer and activator of transcription 3 (STAT3) via the Janus-activated kinase pathway(JAK) [62]. In TNBC, miR-155 is overexpressed and involved in metastasis and poor prognosis by inhibiting von Hippel-Lindau (VHL) tumor suppressor expression and induction of angiogenesis [23]. Moreover, miR-155 induced stemness and decitabine resistance by inhibiting the direct target gene tetraspanin-5 (TSPAN5) in TNBC. This was further confirmed by the overexpression of TSPAN5 that abolished the effect of miR-155 in enhancing stemness and decitabine resistance in TNBC cell lines [63].

### 5.2. Tumor Suppressor miRNAs

Generally, tumor suppressor miRNAs prevent tumors’ development by negatively regulating genes and/or oncogenes that regulate initiation, differentiation, cancer cell progression, or apoptosis. The highly conserved and most abundant miRNA, Let-7c, inhibits cancer cells’ survival by regulating cell proliferation and apoptosis [64], is downregulated in BC cells [65]. MicroRNA-205 is downregulated in TNBC, and its upregulation suppresses invasion and metastasis by inhibiting the expression of ErB3 and VEGF-A in the MDA-MB-231 cell line [66]. Furthermore, over expression of miR-205 can inhibit cell invasion, differentiation, and proliferation by targeting homemobox D10 (HOXD10), which enables to induce P53 and suppress the level of Snail 1 in TNBC cell line [67].

Briefly, dysregulated miRNAs serve as oncogenic or tumor suppressors and may act as potential tools for critical biomarkers in TNBC, which have been involved in cancer stem cell maintenance, epigenetic alteration, apoptosis, proliferation, EMT, invasion, metastasis, prognostic, and radio- and chemotherapy resistance (Figure 1).


## 6. Role of miRNA in Tumorigenesis, Proliferation, and Progression in TNBC

Several studies had already elucidated that miRNAs have played a critical role in tumor cell development, proliferation, migration, invasion, and metastasis in TNBC. Tumor cell invasion is the first step in cancer progression that involves transferring cancerous cells from their origin to neighboring tissues. EMT is the primary mechanism of cancer cells becoming invasive and migratory in which miRNA is involved. In TNBC, miRNAs play a major role in carcinogenesis. However, they have a dual role in tumor-promoting and suppression, oncogenic or oncosuppressor activities. The former involved in the inhibition of endogenous tumor suppressor genes and the later target oncogenes. Altogether, altering miRNA expression is associated with stemness, differentiation, proliferation, autophagy, apoptosis, and several other biological pathways in TNBC [153].

According to previous preclinical and clinical studies, oncogenic miRNAs, such as miR-9, miR-10b, miR-103/107, miR-155, miR-181a, and miR-221/222 [68,69,70,71,72,73,74,75], were upregulated and oncosuppressor miRNAs, including miR-200 family (miR-200c, miR-200b, miR-200b-3p), miR-141, miR-205, miR-199a-5p, miR-3178, miR-212-5p/655, and miR-199/214 cluster [92,105,106,107,108,109,110,111,112,113,114,115,154], were downregulated, resulting in EMT progress promotion by targeting various endogenous molecules. Furthermore, oncogenic miRNAs, such as miR-20a, miR-21, miR-181 family, miR-199a, miR-495, and miR-221/222 [74,76,77,78,79,80,81,155], and tumor suppressor miRNAs, including miR-7, Let-7 family, miR-15b, miR-16, miR-30, mir-33b, miR-34a, miR-103, miR-107, miR-128b, miR-137, miR-145, miR-200 family miR-203, miR-205, miR-223, miR-335, and miR-4319 [116,117,118,119,120,121,122,123,156,157,158,159,160,161,162,163], were involved in the maintenance of cancer cell stemness. Additionally, miRNAs associated with metastasis progress included the overexpressed oncogenic miR-10b, miR-18, miR-21, miR-17/92 cluster, miR-125b, miR-181a, miR-373, miR-455-3p, and miR-629-3p [72,82,83,84,85,86,87,88,89,90,91], as well as under-expressed tumor suppressor Let-7, miR-26a miR-30a, miR-33b, miR-124, miR-126-3p, miR-130a, miR-145, miR-146a-5p, miR-148a, miR-150, miR-200a/b/c, miR-205, miR-206, miR-190a/940, miR-508-3p, miR-519d-3p, and miR-613 [108,124,125,126,127,128,129,130,131,132,133,134,135,136,164,165,166,167,168,169,170,171]. Moreover, miRNAs correlated with promoting proliferation in TNBC include the upregulated oncogenic miR-20a-5p, miR-21, miR-25-3p, miR-135b miR-146a, miR-146b-5p, miR-182, miR-206, miR-498, and miR-502 [77,82,92,93,94,95,96,97,103], as well as downregulated tumor suppressors miR-17-5p, miR-26a, miR-34a, miR-125b, miR-143-3p, miR146a-5p, miR-200c, miR-203, miR-205, miR-211-5p, miR-217, miR-490-3p, miR-539, miR-940 miR-589, and miR-1301 [66,127,135,137,138,139,140,141,142,143,144,145,164,172,173,174,175,176]. Similarly, several miRNAs are associated with anti-apoptotic properties, such as over-expressed miR-17-5p/20a, miR-21, miR-155-5p, miR-182, miR-429, miR-301b, and miR-4458 [77,82,98,99,100,101,102], and also downregulated anti-oncomiR, including miR-10a, miR-31, miR-145, miR-199a-5p, miR-200c, miR-509, miR-890, and miR-1296 [111,146,147,148,149,150,151,152,164]. Comprehensive systematic analysis of existing data showed that oncogenic miR-9, miR-10a, miR-21, miR-29, miR-221/22, and miR-373 were upregulated. Additionally, tumor suppressor miR-145, miR-199a-5p, miR-200 family, miR-203, and miR-205 were downregulated; all are associated with EMT/cancer stem cell (CSC) and invasion in TNBC [153]. Various miRNAs involved in the carcinogenesis of TNBC are summarized in Figure 2.

## 7. TNBC miRNA Therapeutic Targets

Treatment of TNBC patients remains a challenge due to many factors, including the lack of specific target sites, heterogeneity of the disease, rapidly developing resistance to chemotherapy, and limited immunotherapy response. Unlike mutation, epigenetic alterations are reversible in neoplasia, which offers a new opportunity for TNBC treatment. MicroRNAs can be used as anticancer drugs by either increasing or decreasing miRNA levels or improving conventional chemotherapy efficacy in TNBC. Oligonucleotide analogs and antagonists signify the two most common miRNA therapies, but there are also miRNAs involved in increasing tumor cells’ sensitivity to chemotherapy. Below we discussed the pre-clinical and clinical studies regarding these three therapeutic targets that may serve as a potential option in the prevention and treatment of TNBC.

### 7.1. miRNAs Mimics

The downregulation or loss of tumor suppressor miRNA can be overcome by either using ectopic expression of synthetic miRNA mimics [177,178] or utilizing adenovirus-associated transduction vectors of target cells [179,180,181]. MicroRNA analogs are used to restore miRNAs function; this approach is known as miRNA replacement therapy. It has been shown that over-expression of miR-199a-5p was associated with a reduction of cell proliferation, migration, and invasion by altering EMT-related expression of genes including CDH1 and ZEB1 [182], at the same time transfecting miR-199a-5p mimic into BC cell reduced cell proliferation [111]. Micro RNA-125b was suggested to regulate tumorigenesis, tumor progression, poor prognosis, and chemoresistance in TNBC. Studies established on TNBC showed downregulation of miRNA-125b. At the same time, its overexpression promoted the reduction of tumor cell migration and invasion and inhibited EMT in TNBC cell (Hs578T) by targeting mitogen-activated protein kinase kinase 7 (MAP2K7) [143]. It is well known that overexpression of Prxx1 and RASAL2 in BC is associated with tumor cell invasion and migration. Overexpression of miRNA-655 and miR-136 by targeting prxx1 (paired-related homeobox 1) and RASAL2, respectively, suppresses EMT. These two anti-invasive miRNAs are downregulated in TNBC and preventing tumor cell migration and invasion during cancer progression. This study suggests that the mir-655/rxx1 and mir-136/RASAL2/MET axis acts as a suppressor of TNBC metastasis [114,183].

MicroRNA-145 is a tumor suppressor gene that inhibits cancer cell growth, invasion, migration and enhances radio-or chemosensitivity to various cancers using its target site ROCK1. The overexpression of miR-145 downregulates ROCK1 in TNBC, suggesting a potential therapeutic and diagnostic target for TNBC treatment [184]. Similarly, other studies have reported that the cellular inhibitor of apoptosis (clAP1) was suggested as another target for miR-145. Overexpression of clAP1 reduced TNF-α induced apoptosis in MDA-MB-231 cells. At the same time, TNF-α-induced apoptosis was promoted by miR-145 transfection in TNBC, showing that the miR-145-clAP1 axis is a potential therapeutic target site for treating TNBC [150].

Overexpression of miR-200b-3p and miR-429-5p strongly inhibits migration, invasion, and proliferation of TNBC cells by targeting LIM domain kinase 1 (LIMK1) [185], indicating a potential therapeutic target modality of TNBC. Upregulation of miRNA-203 strongly repressed tumor cell proliferation, migration, and invasion in TNBC by targeting baculoviral IAP repeat-containing protein 5 (BRIC5) and Lim and SH3 domain protein 1 (LASP1) [172]. Coronin 1C (CORO1C), an actin-binding protein critical for the control and remodeling of the actin filament network, is increased in TNBC cells, decreasing its mRNA and protein levels by upregulation of miR-206, leading to significantly decreased migration and proliferation [171]. Cyclin D 1 (CCND1), induced by several oncogenic stimuli, plays a vital role in regulating G1-S phase transition and tumorigenesis. This protein is overexpressed in BC and identified as a target of miR-1296 in TNBC, and further confirmed by the over-expression of miR-129 that downregulates the expression of CCND1 and vice versa [151]. BRCA1 is the target of miR-146a and miR-146b-5p associated with proliferation by negatively regulating BRCA1 [93]. In MDA-MB-231 and MDA-MB-468 TNBC cell lines, SOX5 is upregulated while miR-146b-5p is downregulated. MiR-146a-5p as a tumor suppressor is implicated in reducing cell proliferation, migration, invasion, and EMT by targeting SOX5 [127]. CDC27 is a core element of the anaphase-promoting complex (APC) and is associated with controlling mitotic checkpoints to ensure chromosomal integrity, and APC or CDC27 was downregulated in breast cancer. Overexpression of MiR-27a in TNBC by regulating CDC27 or APC in the MDA-MB-231 cell line increased response to radiotherapy [186], indicating that expression of miR-27a might be a promising target for TNBC patients benefited from radiotherapy based therapeutic approach.

### 7.2. miRNA Suppression

Single-stranded oligonucleotides with miRNA matching series are utilized to silence the miRNA function of target proteins. On the other side, miRNA antagonists can suppress the function of miRNAs in TNBC. It is successful in knocking down specific miRNA expression. Locked nucleic acid (LNA) antimiR is another approach. The ribose ring is locked by a methylene bridge and displays unprecedented hybridization affinity towards complementary single-stranded RNA and complementary single double-stranded DNA [187]. In addition to LNA antimiR, miRNAs sponges are used to inhibit miRNA activity competitively. MicroRNA sponges are transcripts that contain multiple and tandem binding sites to a miRNA of interest which inhibits specific miRNA function [188]. Mir-Mask and small-molecule inhibitors are also other strategies to decrease oncogenic miRNAs’ upregulation [189,190]. A common oncogene miR-21 is upregulated in TNBC, and its inhibition reduces the proliferation, viability, and invasiveness and enhances apoptosis in the MDA-MB-468 cell line [191]. Similarly, miR-20a-5p was over-expressed in TNBC and involved promoting migration and invasion by targeting RUNX3 and Bim [192]. A novel target of miR-221, urokinase-type plasminogen activator, which plays an essential role in cell invasion and metastasis via the extracellular matrix’s degradation, contributes to the potential target therapy BC patients [193].

### 7.3. miRNAs Increased Sensitivity to Anticancer Agents

Various studies have reported that miRNAs directly or indirectly enhance cancerous cells’ sensitivity to chemo-or radiotherapy and/or contributed to the delayed chemo- or radioresistance initiation to many cancers, including TNBC. This statement is supported by the evidence that upregulation of miR-451a or miR-130a-3p in TNBC cell MDA-MB-231 strongly enhanced sensitivity to doxorubicin [21] and overexpression of miR-638 increased sensitivity to cisplatin and DNA damaging ultraviolet agents in TNBC [194]. Similarly, up-regulation of miR-200b-3p and miR-190a and low expression of miR-512-5p in TNBC patients are strongly associated with better pathologic response to preoperative chemotherapy [195]. Furthermore, TNBC cell lines (MDA-MB-231-wild type and MDA-MB-436-mutated cells) were treated with gemcitabine alone and combined with PARP1 inhibitor (Poly-ADP-ribose polymerase-1) they revealed an increased expression of miR-26a, -29b, -100, and -148a in MDA-MB-231. Moreover, the addition of PARP1 inhibitor reduced the expression of miR-206 in MDA-MB-231 and increased its expression in MDA-MB-436, suggesting miR-206 may serve as a potential target PARP1 inhibitor [196]. Additionally, overexpression of miR-3613-3p by targeting SMAD2 showed significant inhibition of migration and proliferation and increased sensitivity of the MDA-MB231 cell line to Palbociclib, a selective and potent CDK4 and 6 kinases inhibitor [197].

SIRT1 is a common gene that limits longevity, regulates cellular senescence and serves as a potential target of miR-34 that regulates p53 mediates apoptosis, cell cycle progression, and cellular senescence, where miR-34 inhibits SIRT1 [198]. Similarly, miR-34a and miR-31 targeting the HDAC1/HDAC7-HSP70 K246 axis and protein kinase C epsilon, respectively, increased sensitivity of the DA-MB-231 cell line to anticancer treatment [146,199]. These studies suggest that using mimicking or suppressing miRNA may be an alternative potential therapeutic target in the prevention and treatment of TNBC, and may be used to improve the efficacy of conventional treatment agents (Figure 3).

## 8. Flavonoids: Classes, Subclasses, and Dietary Sources

Naturally-derived products play a significant role in the anticancer drug discovery field, seen by the most frequently used anticancer drug paclitaxel [200]. Flavonoids, plant-based secondary metabolites, are natural products with a polyphenolic structure widely found in different medicinal, nutraceutical, and pharmaceutical applications. They are a large class of compounds ubiquitously found in vegetables, seeds, fruits, cereals, tea, and some beverages, like wine, with anti-inflammatory, antioxidant, anticarcinogenic, and antimutagenic properties [201]. The variety of flavonoids, their widespread distribution, and their low toxicity, relative to other plant active metabolites, guarantee the safety of consuming significant quantities by human beings [202]. Evidence showed that flavonoids impact various cancer processes such as growth, proliferation, differentiation, inflammation, angiogenesis, invasion, and metastasis [203].

### 8.1. Classifications of Flavonoids

More than 5000 different flavonoids are structurally described so far [204]. Flavan is the basic backbone chemical structure of all flavonoids, characterized by two phenolic rings (named A and B) joined by an oxygen-containing heterocycle (C) (Figure 4) [205]. Based on the features of basic structures, flavonoids are classified into six subcategories: flavonols, flavanones, isoflavones, flavan-3-ols (also known as flavanols or cate-chins), flavones, and anthocyanidins (Figure 5) [206].

### 8.2. Dietary Sources of Flavonoids

Since flavonoids are found in many foods and plant-origin beverages, they are termed dietary flavonoids. Flavonoids are the most popular and the largest plant polyphenols found from the everyday plant-source diet. Fruits, vegetables, grains, flowers, stems, roots, bark, tea, and wine are rich natural sources of flavonoids. Studies conducted in 738 men with dietary history to measure the flavonoids quercetin, kaempferol, myricetin, apigenin, and luteolin in foods have shown that tea was the major source that constitute 61% and vegetables and fruits were about 38% of flavonoids [207]. Several previous studies identified and documented the common dietary sources of flavonoids depicted in Figure 6 [208,209,210,211,212].

## 9. Flavonoids in the Prevention and Therapy of TNBC

Anticancer drugs for treating TNBC patients are currently limited since this type of BC lacks the expression of hormone receptors and HER2 amplification [213,214]. Hence, extensive research is directed to the disease’s molecular features, seeking naturally found agents with potential in preventing and treating cancers, including TNBC. Plants (fruits, vegetables, herbs), animals, and microbes, including natural products and secondary metabolites, are sources of many bioactive compounds that have been innovated into drugs to treat disease. Currently, more than 60% of anticancer drugs are obtained from natural products [215]. Active ingredients such as flavonoids, alkaloids, polysaccharides, terpenoids, and saponins derived from natural products have potent anticancer, analgesia, immunomodulation, antioxidant, anti-inflammatory, ant-viral, and antibacterial activities. Since chemoresistance is becoming more frequent and a challenge in treating cancer patients, including TNBC, discovering novel anticancer drugs from natural sources is a priority. Studies have shown that flavonoids have been used in managing TNBC [216]. Moreover, flavonoids also exhibit a strong potential to enhance the effectiveness of currently used conventional chemotherapeutic drugs and, most importantly, they are cost-effective and environmentally friendly [217].

Tumor metastasis is a common feature of TNBC and the major cause of BC-related mortality. In a previous study on the TNBC cell line MDA-MB-231, authors found that Glabridin, an isoflavone from licorice root, strongly inhibited cell metastasis and invasion and decreased tumor angiogenesis [218]. Myricetin-treated MDA-MB-231 cells showed a significant reduction of cell viability, migration, metastasis, invasion, and adhesion through repressing the protein expression of MMP-2/9 [219]. Similarly, studies reported that exposure of the MDA-MB-231 TNBC cell to epigallocatechin-3-gallate, abundantly found in consumed tea, strongly repressed cell proliferation and invasion, caused G0/G1 cell-cycle arrest, reduced activation of MMP-2/9, suppressed expression of Bcl-2/Bax, decreased expression of c-Met receptor and nuclear factor kappa-light-chain-enhancer of activated B cells (NF-kβ) proteins levels, reduced phosphorylation of Akt, and promoted cell death by apoptosis [220]. Furthermore, in vitro studies on breast cancer cases showed that a time-dependent exposure of cells to kaempferol induced G2/M arrest by repressing cyclin A, Cyclin B, and CDK1 as promoted apoptosis by p53 phosphorylation in MDA-MB-231 TNBC cell [221]. Additionally, kaempferol induced caspase-3-dependent apoptotic cell death and reduced tumor growth by inhibiting angiogenic and antiapoptotic gene expressions [222]. Daidzein, one class of isoflavonoid, found in different plants and herbs, has been reported to decrease cell viability, cell migration, and TNBC cell line invasion of MDA-MB-231 cells [223,224,225].

Dysregulation of miRNA is associated with cancer progression and could be an effective target in treating various cancers, including TNBC. Studies have reported that quercetin could modulate oncogenic miRNAs such as Let-7, miR-21, and miR-155, leading to inhibition of cancer initiation, development, proliferation, and apoptosis induction and increased sensitivity of cancer cells to chemotherapy [226]. Furthermore, quercetin inhibits breast cancer cell proliferation and invasion by upregulation of tumor suppressor miR-146a [227]. Likewise, quercetin could modulate the expression of a plethora of miRNAs (48 unique miRNAs). Quercetin strongly repressed tumor metastasis and invasion mediated miRNAs such as miR-146a/b; reduced cell proliferation mediated miRNAs including Let-7 family, and induced apoptosis mediated miRNAs such as miR-605. Quercetin also upregulated tumor suppressor miRNAs such as miR-381 [228]. Exposure of TNBC cells to resveratrol significantly upregulated tumor suppressor miRNAs such as miR-26a in MDA-MB-231 cells [229].

## 10. Molecular Mechanism of Action of Flavonoids in the Prevention and Therapy of TNBC

In general, the molecular mechanism of flavonoids’ action is determined by the compound’s bioavailability at the target tissue in the body. Studies demonstrated that the concentrations of flavonoids and their metabolite byproducts found in vivo are relatively lower compared to other nutrients [230]. Flavonoids undergo extensive metabolism in the large and small intestine, while a great portion of flavonoids that are not absorbed in the intestine reach the colon, where the microbiota catabolizes the unabsorbed flavonoids into smaller molecules, which facilitates the absorption and enhance the bioavailability [231]. Interestingly, modification of the flavonoid chemical structure and conjugation with metal ions significantly enhance the bioavailability of flavonoids [217].

Although most flavonoids are considered safe, excessive consumption may be associated with adverse health effects, including gastrointestinal symptoms, allergy, anemia, and hepatotoxicity [232]. Following consumption, flavonoids have shown preventive or curative effects in different diseases, including TNBC, by modulating various cellular pathways involved in developing various diseases [233].

Extensive studies showed that flavonoids and their metabolites exert modulatory actions in cells through impacting—stimulating or inhibiting—several signaling pathways. These natural products affect the expression of genes. They may be associated with cellular growth, proliferation, angiogenesis, migration, invasion, differentiation, apoptosis, mitochondrial dysregulation, alteration of the cell cycle, induction, or inhibition of autophagy, and affecting downstream signaling transduction. In brief, we have discussed the modulatory effects of several flavonoids that could be attractive targets for drug development to be used in the prevention and therapy of TNBC.

### 10.1. Mitochondrial Regulation

Mitochondria is the powerhouse of the cell that mainly produces ATP. Still, it is also involved in the generation of ROS, redox molecules and metabolites, regulation of biosynthetic metabolism, and cell signaling and cell death. Evidence demonstrated that impaired mitochondrial function is found in TNBC cells due to switching to higher glycolysis rates and producing a high lactate amount. Observed lower expiration rate, associated with reduced expression of complexes I and III protein components and increased Akt activation, further promotes glycolysis. In TNBC cells, this high glucose demand enables the tumor to provide the energy essential for proliferation. Therefore, TNBC is very sensitive to glycolytic inhibitor agents compared to positive estrogen cells [234]. Furthermore, loss of p53 could impair the mitochondrial respiratory chain activity and enhance the shift from oxidative phosphorylation to glycolysis [235]. The most well-studied flavonoid, luteolin, enhances the tumor suppressor P53 and increases apoptosis by upregulation of miR-34a-5p and downregulation of miR-21 [236,237,238], indicating that repression of p53 through regulating miRNAs (miR-34a-5p/21) could reverse the impaired mitochondrial respiratory chain activity.

Apoptosis is an attractive target in cancer therapy. It is frequently produced by mitochondrial dysfunction, by opening a non-specific mitochondrial permeability transition pore in the inner mitochondrial membrane to release the mitochondrial components into the cytoplasm. One of these components, ROS, is the key inducer of apoptosis [239,240], as revealed by previous studies on the MDA-MB-231 TNBC cell line. This study reported that overexpression of miR-223 dysregulated the mitochondria/ROS pathway, ultimately inducing apoptosis and activating effector caspases 3, 7, and 9 [121]. Moreover, studies proved that a myricetin-derived flavonoid, oncamex, reduced cell viability and promoted cytotoxicity and apoptosis and increased caspase activation in a mitochondrial-dependent pathway of MDA-MB-231 and other TNBC cell lines [241]. Therefore, the flavonoid role in regulating miRNAs might be a promising agent in controlling mitochondrial respiratory chain activity and promoting apoptosis in the mitochondrial-dependent pathway.

### 10.2. Induction of Apoptosis

Compounds which induce apoptosis represent a significant part in the treatment and prevention of tumor. In cancer pathogenesis, evasion of apoptosis is one of the mechanisms that has been utilized by cancer cells. Therefore, apoptosis induction is considered one of the novel strategies for anticancer drug development. Indeed, many tumor suppressor miRNAs are involved in the induction of apoptosis in TNBC through regulating various endogenous signaling molecules associated with apoptosis [146,147,148]. This impact was exhibited by several flavonoids.

Fisetin, a class of flavones, induces apoptosis in MDA-MB231 and MDA-MB-468 TNBC cells through inhibition of Aurora B kinase and initiator activation caspases as well as effector caspase target PARP1 molecules. Similarly, exposure of the MDA-MB-231 TNBC cell line to fisetin showed a reduction in cell division and promoted apoptosis through activation of initiator caspases 8 and 9 and poly ADP-ribose polymerase-1 cleavage. In TNBC cells, the noticeable reduction in apoptosis upon treating cells with caspase inhibitor suggested the caspase-dependent apoptosis induction mechanism in this subtype of BC8 [242].

Myricetin, another class of flavonoids, induced apoptosis in TNBC through intracellular ROS induction, promoting activation of extracellular regulated kinases 1/2 and p38 mitogen-activated protein kinase mitochondrial membrane disturbance, cytochrome C release, and double-strand DNA breaks. This study was further confirmed by using the antioxidant N-acetyl cysteine, which reverses the myricetin-induced cytotoxicity due to DNA damage and the inhibition of intracellular and extracellular accumulation ROS using deferiprone and superoxide dismutase and catalase, respectively [243].

The flavonoid artonin E also has been found to induce apoptosis in a concentration and time-dependent manner by increasing both caspase 8 and caspase 9 and intracellular total ROS in TNBC cell MDA-MB-231. Furthermore, this study showed that artonin E inhibits the expression of anti-apoptotic proteins and apoptosis inhibitors such as HSP70, Bcl-x, livin, and P53, which are aggressively overexpressed in TNBC cells [244].

In modulating miRNAs, the flavonoid class xanthomicrol promotes apoptosis and suppresses angiogenesis by upregulating tumor suppressor miRNAs, such as miR-29 and miR-34 suppression of oncomiR, including miR-21, miR-27, and miR-125 in TNBC [245].

### 10.3. Induction of Cell-Cycle Arrest

Several mechanisms are controlling the processes of the cell cycle to ensure proper cell division. For instance, cyclin-dependent kinases (CDKs) regulated by cyclins are involved in the mediating cell cycle. Disturbance of the cell cycle due to dysregulated CDKs has been one of the main hallmarks of many cancers, associated with induction of unscheduled cell proliferation as well as chromosomal and genomic instability [246,247].

PTEN acts as a tumor suppressor with the potential to induce cell-cycle arrest by negatively regulating PI3K/Akt signaling pathway. Previous studies on the MDA-MB-231 cell line have shown the implication of miR-498 in the elevated cell proliferation and cell cycle progression by suppressing PTEN and activating PI3K/Akt signaling [248]. On the contrary, miR26b was found to increase G0/G1 cell-cycle arrest and reduce cellular proliferation through targeting CDK8 in the MDA-MB-231 TNBC cell line [249].

Through regulation of miRNAs, Flavonoids have been shown to enhance cell-cycle arrest in many cancers, including TNBC. It has been reported that fisetin-treated TNBC cells showed fewer cells in the G1 phase of the cell cycle, while an increased percentage of G2/M phase cells was detected, suggesting the induction of G2/M phase arrest [242]. Similarly, artonin E arrested the cell cycle at the G2/M phase in TNBC. Incubating MDA-MB-231 TNBC cells with artonin E showed a significant increase in the percentage of cells at the G2/M phase compared to the control while accumulating cells in the G0/G1 phases, indicating the existence of necrotic cells [244]. Moreover, the flavonoid xanthomicrol increased the induction of cell-cycle arrest (G1-arrest) and apoptosis by downregulating oncogenic and upregulating tumor suppressor miRNAs, leading to shrinkage of TNBC tumors in mice [245].

### 10.4. Induction of Autophagy

Autophagy, also known as self-eating, is a lysosomal degradation process found at low levels under normal physiological conditions to maintain cellular homeostasis by balancing catabolic and biosynthetic processes. Still, it is induced in response to stressful conditions for removing damaged proteins and organelles [250]. Autophagy involves five critical steps: induction, nucleation, vesicle lengthening and maturation, vesicle fusion, and, lastly, degradation and recycling. These vital processes are controlled by recruiting an autophagy-related protein (ATG) repressed by mTOR1 (mechanistic target of rapamycin complex1). In cancer cells, autophagy is a “double-edged sword.” At the early stage of the disease, this mechanism acts as a tumor suppressor by degrading oncogenic protein, reducing inflammation and chronic tissue damage, preventing mutations and genetic instability; however, late-stage cancer uses autophagy for survival cellular stress situations [204,247,248].

Many studies have shown that a high autophagy level is associated with chemo-resistant and maintenance of aggressive tumor behavior in TNBC [251]. For instance, authors reported that the yes-associated protein’s pivotal element in promoting autophagy is upregulated in TNBC cells compared to hormone-positive breast cancer cells [252]. Similarly, treating MDA-MB-231 TNBC cells with the autophagy inhibitor chloroquine enhanced the cytotoxicity potency of drugs and inhibited the autophagic flux [253]. On the contrary, modulation of autophagy was reported to stimulate the antitumor immune response. Autophagy deficiency promoted TNBC resistance to T cell-mediated cytotoxicity by blocking tenascin-C degradation, suggesting an important immunosuppressive factor regulating the activity and infiltration of cytotoxic T cells in TNBC [254]. Several miRNAs, including miR-7, 21, 25, 99a, 338, 375, 382, 290-295, 133a-3p, 244-3p and miR-409-3p, regulate autophagy through modulation of various signaling pathways, such as PI3K/Akt/mTOR [255].

Evidence has shown that flavonoids induce autophagy and apoptosis and inhibit cell proliferation through downregulation of the PI3K-mediated PI3K/AKT/mTOR signaling pathway in human breast cancer, including MDA-MB-231 TNBC cells [256]. In parallel, apigenin induces autophagy and apoptosis simultaneously in MDA-MB-231 cells by inhibiting the PI3K/AKT/mTOR pathway [257].

### 10.5. Inhibition of Cell Migration, Metastasis, Invasion, and EMT

EMT endorses the tumor cells to detach from the original cancer site and travel into the distant normal tissue, blood, and lymphatic system to form metastatic lesions [258]. Normal cells undergo an apoptotic process, known as anoikis, immediately after losing contact with an extracellular matrix or neighboring cells [259].

Increasing evidence indicates that many miRNAs are involved in tumor cell migration, metastasis, invasion, and EMT. Authors have reported tumor inducer miRNA, let-7g, miR-21, miR-101, miR-125a-5p, miR-424, miR-579, and miR-627 are strongly involved in the induction of EMT, cell migration, metastasis, and invasion in TNBC [191,260]. Similarly, in TNBC, miR-655 is downregulated, and its overexpression strongly reduced EMT. Emerging evidence confirmed that ectopic expression of miR-655 in MDA-MB-231cells induced up-regulation of cytokeratin, downregulated the expression of vimentin, and repressed migration and invasion mesenchymal-like cancer cells suggested that miR-655 inhibits EMT through downregulation of prrx1 [114].

Studies have reported that flavonoids reduced tumor cell migration, metastasis, and invasion in TNBC. Fisetin, a class flavonoid, has been described to possess anticancer activity in different cancers, exhibited suppression of cancer cell growth, invasion, migration, promoted cell-cycle arrest, autophagy mechanism, and apoptosis [261]. Also, increasing evidence has demonstrated that fisetin significantly reduced cell proliferation, migration, and invasion and reversed the EMT process in a concentration-dependent manner [262]. Similarly, quercetin significantly downregulated tumor metastasis and invasion miRNAs such as miR-146a/b, miR-503, and miR-194 [228].

### 10.6. Enhancement of Immune Responses

The entire immune system plays a crucial role in preventing cancer incidence and destroying tumors, with no toxicity to normal cells. Both adaptive and innate immune effector mechanisms could be tools in identifying and controlling cancer cells. NK cells are the ones that detect the first transformed cells via their counter with particular ligands on tumor cells, which then leads to the removal of transformed cells. Activated macrophages and dendritic cells are involved in the removal of transformed fragment cancer cells. Many inflammatory cytokines are secreted and present tumor cell-derived molecules to B and T cells upon activation of macrophages and dendritic cells.


Furthermore, these two cells enhance the production of additional cytokines to activate innate immunity and support the secretion and expansion of tumor-specific T cells and antibodies [263,264]. The subpopulation of tumor-infiltrating lymphocytes, CD8+ T cell infiltration, the T cell regulatory molecule, and the programmed cell death protein (PD-L1) are highly expressed in TNBC; approximately 20% of TNBC tumors express PD-L1 [265]. Cytotoxic T-lymphocyte-associated antigen and PD-L1 are the two immune checkpoints targeted by anticancer drugs to augment antitumor immunity [265,266]. 

A comprehensive systematic review of previously published data showed that miRNAs regulate the immune response in several ways. This analysis explained miRNAs’ role, such as miR-146a, miR-155, miR-181a, and miR-223, during immune system activation. For instance, miR-125b was strongly expressed in human immature CD4+ T cells, regulating the expression of IL-10 receptor-α, IL-2 receptor-β, and IFN-γ, and it is downregulated during differentiation effector T cell subsets. Furthermore, from this reviewed data, due to T cell development and differentiation alteration, miR-20a-5p, miR-24-3p, miR-106a-5p, and miR-891a are downregulated, suggesting that tumor cells can make alterations of the immune system to grow and metastasize [267]. In line with this, another extensive reviewed analysis demonstrated that immune-modulatory miRNAs, including miR-19a-3p, miR-23/a/24-2/27a, miR-29c, miR-101, and miR-126/126a, are down-regulated, whereas miR-155, miR-181-1b, miR-223, and miR-494 were upregulated in various types of cancers, including breast cancer [268].

Flavonoids such as quercetin, luteolin, apigenin, and fisetin could reverse the impaired immune system response by repressing inflammatory cytokines, reducing activated dendritic cells and proliferation of T cells, inhibiting antigen-specific production of IFN-gamma, reducing mast cell activation and T-helper cell cytokine activation, decreasing T CD4+, T CD8+ and eosinophils, suggesting another promising potential target site in the treatment of various cancers, including TNBC [269]. In consistency with this evidence, the dietary apigenin significantly decreased LPS-induced expression of miR-155, resulting in restoring the impaired immune balance of mice during inflammation [270].

### 10.7. Promotion of Differentiation

Differentiation therapy is one type of promising cancer stem-cell-targeting therapy in BC, a source of breast tumors including TNBC. Differentiation aims to attack the stemness of cancer stem cells, leading to the reduction of their chemo-and radioresistance. The main objective of differentiation therapy is to induce differentiation of tumor cells, consequently stopping proliferation, and ultimately control tumorigenic and malignant potential [271].

MicroRNAs such as miR-1 and miR-206 promote differentiation in different cancers. In particular, tissue-specific tumor-suppressing miRNAs can enhance tumor cells’ differentiation to their original counterparts and solid malignancies to their normal tissue types [272]. Cumulative evidence has shown that differentiation therapy possesses a low toxicity profile [273]. Exposure of MDA-MB-231 to arsenite reduces tumor volume and weight in mouse xenografts due to the induction of differentiation [274,275]. 

Flavonoids cause undifferentiated cancer cell lines to differentiate into cells, demonstrating mature phenotypic characteristics. Authors reported that many flavonoids were found to promote the differentiation of tumor cells in various cancers. Induction and promotion of differentiation by flavonoids may lead to an ultimate removal of tumorigenic cells and rebalance regular cellular homeostasis [276].

### 10.8. Inhibition of Proliferation

Dysregulation of proliferation appears to be the main hallmark of susceptibility to tumors [277]. Potent oncogenes, miR-21 and miR-191, are associated with increased tumor cell proliferation and invasion in the MDA-MB-468 TNBC cell line [191,278]. In preventing cancer, reversion, inhibition, or hindering cellular hyperproliferation is generally the main central point.

Plenty of studies have shown that most flavonoids have been demonstrated to suppress proliferation in several human cancers, including TNBC [279]. It is proposed that the possible mechanistic action of flavonoids is through inhibiting the prooxidant process that causes tumorigenesis. The tumor promoter, xanthine oxidase, may activate prooxidant enzymes, and at the same time, flavonoids could inhibit polyamine biosynthesis activated by xanthine oxidase. Moreover, flavonoids could inhibit polyamine biosynthesis catalyzed by its rate-limiting enzyme ornithine decarboxylase, which is associated with DNA synthesis and cell proliferation in many tissues. Additionally, flavonoids are involved in the repression of signal transduction enzymes such as protein tyrosine kinase (PTK), phosphoinositide-3-kinases (PIP3), and protein kinase C (PKC), which are significantly involved in the regulation of cellular proliferation [280,281,282]. Similarly, 3,6-dihydroxyflavone suppresses proliferation by upregulating tumor suppressor miR-34a and downregulating the oncogenic miR-21 in breast cancer cells, such as MDA-MB-231 [283]. Additionally, curcumin upregulation of the tumor suppressor miR-34a ultimately suppresses TNBC cell proliferation [284].

### 10.9. Inhibition of Oxidative and Nitrosative Stress

Cancer cells grow in hypoxia and adapt their metabolism to fulfill the nutrients and energy required for proliferation and survival. In response to low oxygen concentration, elevated levels of reactive oxygen and nitrogen species (RONS) are observed in various cancer cells, including TNBC [285]. Cellular redox homeostasis is maintained under normal physiological conditions by balancing the endogenous antioxidant system and RONS generation. Dysregulation in this balance has been associated with cell proliferation, differentiation, angiogenesis, evasion of apoptosis, migration, and survival, with the tendency to introduce and promote tumorigenesis [286,287,288,289].

MicroRNAs such as miR-21, miR-miR-28, Mir-93, miR-144,miR-155, and miR-200a, miR-212 could regulate RONS -antioxidant balance by targeting Nrf2/Keapl, mitochondrial, SOD/catalase pathways that affect carcinogenic process [290]. Furthermore, elevated RONS could also induce abnormal expression of specific miRNAs. Overexpression of RONS could result in upregulation of miR-21 and downregulation of miR-27, miR-29b, and miR-328, suggesting that miRNAs are RONS sensitive [291].

Dietary flavonoids are natural antioxidants with the one-electron donor that could act against cancer by restricting the damage of oxidative reactions in cells, enhancing cancer growth. Flavonoids serve as antioxidants by scavenging singlet oxygen, superoxide anion, lipid peroxy-radicals, and /or balancing free radicals involved in the process of oxidation through complexing with oxidizing species or hydrogenation [286,292]. Additionally, flavonoids are involved in the downregulation of the oncogenic miR-21, contributing to the overexpression of RONS [283]. Hence, increased oxidative and nitrosative stress markers may serve as prognostic predictors and potential targets for therapeutic and/or preventive strategies in TNBC.

### 10.10. Reduction of Angiogenesis

Angiogenesis is a strictly regulated process in conventional physiological activity controlled by different endogenous angiostatic and angiogenic factors. However, disease conditions, such as cancer, affect these factors and lead to pathological angiogenesis [293]. In the tumor, angiogenesis enhances the proliferation of existing blood vessels entering the cancer cells, to provide oxygen and nutrients and remove any metabolic waste products from the tumor. Hence, it is essential for the development, aggression, invasion, and metastasis of solid tumors [294]. Several proangiogenic factors are involved in the angiogenesis process, including vascular endothelial growth factor, interleukin 6/8, fibroblast growth factor, transforming growth factor-alpha, and prostaglandin E. Among these, VEGF is the key regulator proangiogenic element that regulates angiogenesis. Therefore, VEGF inhibition is a promising target for the prevention and therapy of cancer [295,296].

Emerging evidence has shown that miRNAs are associated with the process of angiogenesis through regulating the expression levels of anti-or-pro angiogenic factors. For instance, a member of the miR-106b~25, miR-93-5p, is up-regulated in various cancer types, including TNBC, and promotes tumor angiogenesis [76].

Studies both in vitro and in vivo showed that wogonoside, a bioactive flavonoid, effectively inhibited angiogenesis in TNBC by decreasing the expression of VEGF in MDA-MB-231 and MDA-MB-468 cells [297]. Additionally, resveratrol was previously found to down-regulates the expression of miR-93-5p in breast cancer cells [298], suggesting the importance of flavonoids in decreasing the nutritional source of tumor cells and ultimately leading to death.

### 10.11. Modulation of Signaling Transduction

The conversion of external stimuli into intracellular signals mediates gene transcription, activating cellular machinery appropriately and has a pivotal role in the normal cellular signaling process. The systemic function of sequential cascades maintains the physiological process’ homeostasis, while their abnormal activation results in impaired cell proliferation, differentiation, and apoptosis. Some of the mechanisms of flavonoids used in the prevention and treatment of TNBC through modulation of signaling transduction are the protein kinase C (PKC), mitogen-activated protein kinase/extracellular signal-regulated kinases (MAPK/ERK), phosphatidylinositol 3-kinase (PI3K)/Akt, wnt/β-catenin, JAK/STAT3, Nrf2/keap1, NF-kβ, and SMAD2/3. Knowingly various signaling molecules can exert similar effects on the same cell type, and at the same time, individual signaling molecules can have numerous physiological functions. The complexity of function because of overlapping downstream signaling pathways attracts attention to appropriately identifying each signaling molecule’s critical role and their downstream cascades. Plenty of studies have demonstrated the interference between flavonoids and these signaling transduction pathways. The elucidation of the cellular and molecular events affected by flavonoids reveals novel strategies and molecular targets for developing similar acting compounds used for the prevention and treatment of TNBC.

#### 10.11.1. PKC Pathway

PKC is a well-known integral part of the cell signaling machinery enzyme. Even though PKC activation is critical for the signal transduction pathways, the homeostatic regulation of PKC activation is vital for normal physiological cell function. Meanwhile, this enzyme’s unusual continuous stimulation may lead to uncontrollable cell growth and proliferation [299]. PKC theta and alpha isoforms are mainly upregulated in TNBC cells and enhance the growth, proliferation, EMT, survival, migration, and tumor cell invasion. However, all of these effects are reversed by PKC inhibitors [300,301].

Emerging evidence showed that the PKC alpha subtype is detected in advanced-grade TNBC cells. On the other side, miR-200b functions as a tumor suppressor to reduce cell migration and invasion by targeting PKCα in TNBC cell lines such as MDA-MB-468 [108]. Previous studies using both in vitro and in vivo models revealed that quercetin has antiproliferative activity through inhibiting signal transduction targets such as PKC [302]. These data might suggest that the miRNAs-dependent activation of PKCs is a promising target for several drugs, including flavonoids.

#### 10.11.2. Nrf2/keap1 Axis

Nuclear factor erythroid 2-related factor 2 (Nrf2) is a transcription factor that regulates cellular stress caused by electrophiles, xenobiotics, and oxidants. Nrf2 provides cytoprotective activities, repressing oxidative stress response levels in many cancer cells. To be activated, Nrf2 must be detached from an inhibitor Kelch-like ECH-associated protein 1 (Keap1) and shuttles to the nucleus to bind with the target gene and enhance gene expression, since, under homeostatic conditions, Nrf2 interacts with Keap1 in the cytosol [303]. The cancer-preventive role of Nrf2 has been extensively studied. Nrf2 prevents tumorigenesis by quenching ROS, ensuring the quick enzymatic modification and excretion of carcinogenic chemicals, or repairing oxidative-related damage via target gene expression [304]. However, recent findings describe the “dark side” of Nrf2, as its activation is involved in promoting metastasis, increasing cancer progression, and conferring resistance to radiotherapy and chemotherapy [305,306]. Studies showed that overexpression of tumor suppressor miR-29b-1-5p inhibits Nrf2 in the MDA-MB-231 TNBC cell line [307].

Flavonoids such as chrysin [308], luteolin [309], and apigenin [310] have exhibited antitumor activity by strongly reducing Nrf2 mRNA through various mechanisms, such as down regulation of the PI3K/Akt pathway, decreasing antioxidant gene expressions, inhibiting proliferation, invasion, and migration of tumor cells, as well as increasing sensitivity to anticancer drugs. Moreover, flavonoids such as epigallocatechin-3-gallate upregulate tumor suppressor miR-29b-1-5p [311] could repress the expression of Nrf2. A comprehensive review of existing data showed that flavonoids such as thymoquinone and epigallocatechin increase activation of Nrf2, indicating a cytoprotective effect against several diseases, including TNBC [312].

#### 10.11.3. PI3K/Akt/mTOR Signaling Pathway

PI3K/Akt is one of the most pro-survival intracellular signaling systems [313]. PI3K activation and its downstream effector, Akt, have been shown to repress apoptosis and promote cells’ survival by enhancing anti-apoptotic proteins’ expression and reducing pro-apoptotic proteins activity. Additionally, PI3K/Akt inhibition abolishes cell survival and promotes apoptosis [313,314,315]. The PI3K/Akt/mTOR pathway is positively associated with cell proliferation and survival, and any dysregulation of this signaling pathway significantly mediated the progression of various cancers, including TNBC [316]. PI3Ks are enzymes activated by receptor tyrosine kinases and non-receptor tyrosine kinases that mediate cellular signal transduction by phosphorylating membrane inositol lipids and generating the secondary messenger, PIP3 (phosphatidylinositol 3,4,5- triphosphate), which in turn drives a conformational change in Akt/PKβ, leading to activation of mTOR. In addition, mTOR involves induction of cell survival, cell cycle progression and proliferation, angiogenesis, and migration [317]. The tumor suppressor PTEN is negatively regulating Akt via dephosphorylation of PIP3, which is essential for Akt activation, is mostly deleted, mutated, or reduced in TNBC. MicroRNA-184 modulates the PI3K/Akt/mTOR signaling pathway [318] and miR-124, a potential oncogenic in breast cancer targets PTEN-PI3K/Akt pathway [319]. These studies suggested that the PI3K/Akt/mTOR pathway is a promising chemotherapeutic target in treating TNBC and other cancers.

Flavonoids modulate the cell survival signaling pathway through its interaction with PI3K/Akt [320]. Quercetin, a class of flavonoids, acts as an Akt inhibitor that suppresses tumor cell survival, migration, and metastasis. Moreover, upregulation of PI3K/Akt is associated with a decrease in docetaxel effectiveness, and in combination with quercetin, the response rate was enhanced. This study suggested that quercetin improves chemotherapy’s efficacy and increases the accumulation of drugs in the tumor site through inhibition of the Akt/MMP-9 pathway, indicating a promising potential target in treating metastatic TNBC [321].

#### 10.11.4. MAPK/ERK Signaling Pathway

MAPK is a family of distinct signaling cascades in the cell and acts as a central point in reaction to different extracellular stimuli [322]. MAPKs regulate the expression of genes that control inflammation, proliferation, cell survival, inducible nitric oxide synthase, cytokine expression, and collagenase production [323]. MAPKs have three main classes: ERK, c-Jun N-terminal kinase(JNK), and P38 [324]. MAPK/ERK is the most determinant of cell differentiation, cell growth, cell survival, and motility via activation of the cAMP response element-binding protein, upregulation of anti-apoptotic protein Bcl-2, and inhibition of Bcl-Xl/Bcl-2 associated with death promotion [325]. The MAPK/ERK signaling pathway is significantly activated in TNBC, and the high protein expression levels correlate with a shorter survival rate in TNBC patients [326].

MAPK pathways are characterized by various key genes, such as Ras, Raf, MEK, and ERK. Notably, activation of Ras results in Raf phosphorylation, thereby enhancing MEK and ERK activation, and ultimately leads to tumor cell proliferation and cell survival. These cascades are significantly regulated by various miRNAs described else-where [327]. Studies have reported that downregulation of miR-489 was associated with overexpressed MAPK signaling pathways. A significant reduction in cell growth and tumorigenicity was found in BC with upregulated expression of miR-489 [328].

Emerging evidence has shown that myricetin protects oxidative stress-induced cytotoxicity by inhibiting the P38/MAPK/JNK/ERK signaling pathway. The finding was further confirmed by the addition of p38 MAPK inhibitor and resumed the protective effect of myricetin [329].

#### 10.11.5. NF-Kβ Signaling Pathway

NF-kβ is an essential transcription factor for regulating apoptosis, cell proliferation, and several pro-inflammatory proteins such as chemokines, cytokines, prostaglandins, nitric oxide, leukotrienes, and adhesion molecules [330]. Any dysregulation in the activation of NF-kβ enhances cancerous cells’ survival and contributes to an increment of chemotherapy resistance [331].

NF-kβ activation involves induction of the EMT process, increased CXCR-4 expression, promotion of metastasis, and enhancement of chemotherapy resistance [332]. Tumor cells also enhance NF-kβ activity by enhancing cytokine release from stromal cells and fibroblasts in the tumor microenvironment. Furthermore, many external stimulators like ROS and hypoxia-induced factors can increase NF-kβ activation [333,334,335]. In TNBC, overexpression of NF-kβ has been significantly demonstrated [336]. Several studies elucidated the main consequences of NF-kβ activation, including enhancing invasiveness of tumors, promoting cell detachment, increasing neoangiogenesis, upregulating MMPs expression, inducing tumor proliferation, polarization of tumor cells, production, and overexpression of inflammatory reparative response, which is ultimately causing further development of tumors [337].

A comprehensive review of existing data showed that miR-146 negatively regulates the expression of NF-Kβ, exhibited an essential role in reducing tumorigenesis and progression by suppressing tumor cell migration and invasion. Similarly, upregulation of miR-21 and miR-301a activates NF-Kβ in various types of cancers, including TNBC [338].

Flavonoids like fisetin inhibit the activation of the NF-kβ signaling pathway, leading to down-regulation of gene products that decrease apoptosis and enhance tumor cells’ metastasis [339,340]. Similarly, treating MDA-MB-231 TNBC cells with flavonoids were previously found to induce significant inhibition in NF-kβ signaling through different genes regulating this pathway, as revealed by the downregulation of cyclooxygenase-2 (COX-2) and MMP9 [341]. In MDA-MB-231 cells, flavonoids potentially suppress carcinogenesis through overexpressing tumor suppressor miR-34a and repressing the oncogenic miR-21 [283], indicating that altering miRNAs by flavonoids might be essential to prevent NF-Kβ signaling-dependent induction of tumorigenesis in TNBC.

#### 10.11.6. JAK2/STAT3 Signaling Pathway

Signal transducer and activator of transcription (STAT) is a common transcription factor that binds to DNA and induces various gene expressions. There are many isoforms of STAT, from STAT 1–3. However, STAT3 is the most studied sub-family that binds to DNA in response to cytokines (IL-6) and epidermal growth factor, and is significantly associated with cancer development, proliferation, migration, angiogenesis, metastasis, immune suppression, poor prognosis, apoptosis, and chemoresistance in TNBC [342]. Activation of the JAK2/STAT3 pathway is evaluated by phosphorylated STAT3 using a kinase protein JAK. Inhibiting this pathway impairs tumor growth and metastasis, indicating an effective therapeutic target in TNBC [343,344].

Upregulation of oncogenic miRNAs, such as miR-18a, miR-30, miR-155, and miR-221, leads to elevated JAK2/STAT3 expression, ultimately resulting in tumorigenesis [345]. Due to its involvement across various types of cancers, including TNBC, the JAK/STAT pathway is a potentially attractive therapeutic target in the prevention and treatment of TNBC. It has been reported that dietary flavonoids luteolin and quercetin inhibit migration and invasion of cancer cells by reducing the JAK/STAT signaling molecule [346].

#### 10.11.7. Wnt/β-Catenin Signaling Pathway

Wnt, a highly conserved group of secreted glycoproteins, correlates with several cellular functions, such as stem cell renewal, organ formation, and cell survival, and is secreted by cells into the extracellular space to activate receptor-mediated signaling in the immediate vicinity [347].

The Wnt pathway’s ability to trigger different intracellular signaling pathways highlighted its importance as one of the essential molecular cell-to-cell signaling mechanisms. It is noteworthy that this pathway with adequate regulation mechanisms is essential in physiological conditions. A classical signaling pathway in the Wnt signaling route is a β-catenin-mediated path, in which β-catenin plays a role in preventing the movement of cells by forming a complex with E-cadherin. Augmentation of Wnt signaling enables the accumulation of β-catenin in the cytoplasm and transfers into the nucleus, leading to activation of proto-oncogenic genes like cym c cy-clinD1, and eventually, cancer initiation and development [348].

Mutations in the component of Wnt/β-catenin could also endorse the inappropriate regulation of this pathway, are frequently associated with many cancers, and induce tumor recurrence [349,350]. Wnt/β-catenin signaling super-activation is positively correlated with high grade, metastatic, and poor prognosis in TNBC cells [351,352]. Studies have been reported that upregulation of Wnt/β-catenin was shown to be one of the mechanisms to resist PI3k inhibitors. In addition, the use of β-catenin inhibitors sensitizes breast cancer cells to PI3K inhibitors [353].

Emerging evidence showed that upregulation of miR-21 enhances cancer cell proliferation and metastasis through a significant activation of the Wnt/β-catenin signaling pathway [354]. Similarly, miR-301a strongly induces Wnt/β-catenin signaling in breast cancer cells by targeting PTEN, a master regulator of several oncogenic pathways such as PI3K/Akt [355]. The role of oncogenic and tumor suppressor miRNAs in controlling the activation of the Wnt/β-catenin signaling pathway was described elsewhere [356].

Studies demonstrate that flavonoids such as quercetin, apigenin, and fisetin could interact with β-catenin and accelerate its degradation and disruption of Wnt/β-catenin signaling in many cancers, including breast cancer [357]. Similarly, treating cancer cells with genistein can repress the Wnt/β-signaling pathway through downregulation of the oncogenic miR-1260b [358].

#### 10.11.8. SMAD Signaling Pathway

Smooth-muscle-actin and MAD-related proteins (SMAD), which have seven major classes of SMAD proteins (SMAD 1-7), is a family of proteins, a component of an evolutionarily conserved pathway in humans. SMAD is a crucial downstream of TGF-β signaling regulating the transcriptional response for TGF-β functions [357]. SMAD family member 2 (SMAD2) is overexpressed in BC and promotes tumor progression [356].

It has been reported that inhibitors that reduce SMAD2 and SMAD3 by interrupting the TGF-β signaling pathway could help treat and prevent human cancers, including TNBC. Indeed, miRNAs such as miR-27a and miR-3613-3p were significantly reduced in TNBC patients. Recent studies have demonstrated that miR-27a by regulating SMAD2 inhibits tumor growth and migration by interrupting TGF-β signaling, ultimately leading to inhibition of cancer cell proliferation, induction of apoptosis, and decreased tumor cell migration [357]. Similarly, overexpression of miR-3613-3p decreased tumor proliferation and migration in TNBC cells via targeting SMAD2 and EZH2, EZH2 like SMAD2/3, significantly promoting cancer cell proliferation in many cancer cells, including breast cancer [197]. Flavonoids like curcumin upregulate miR-27a in various types of cancer, including breast cancer, leading to reduced carcinogenesis [358].

The modulation of signaling transduction pathways by flavonoids is summarized in Table 1.

Taken together, dietary flavonoids have shown various mechanistic actions that might be useful in the prevention and treatment of TNBC, as summarized in Figure 7.

## 11. Conclusions

The complexity of signal networks driving tumorigenesis in TNBC, with the absence of a specific target for treatment, contributes to a poor clinical outcome in patients. Ongoing and completed preclinical and clinical studies have examined and reported flavonoids’ efficacy and safety as anti-cancer agents. Flavonoids are generally nontoxic and demonstrate a broad-spectrum range of advantageous physiological activities. Numerous studies reported and widely discussed the role of dietary flavonoids in cancer prevention and cancer, including TNBC. Since they are widely distributed in fruits and vegetables, many in vitro and in vivo studies have generated data that high dietary intake of flavonoids could be associated with a low prevalence of cancer in humans. Flavonoids have been shown to have effects such as: Antiproliferation, anti-angiogenesis, regulation of mitochondrial and autophagy activities, carcinogen in-activation, promotion of apoptosis, antioxidation, promotion of differentiation, regulation of different signaling pathways, augmented chemotherapeutic drugs, and reversal of multidrug resistance. These promising effects of flavonoids are utilized as the basis drug development from dietary flavonoids to prevent and treat TNBC.


MicroRNAs are largely dysregulated and expressed in most cancers, including TNBC which has been associated with tumor initiation and progression, and represent a desirable and potential target for the treatment and prevention of TNBC. Flavonoids display a unique ability to change miRNAs’ levels via different mechanisms, including transcriptional, epigenetic, and miRNA processing, downregulating oncogenic miRNAs, and upregulating tumor suppressor miRNAs. Additionally, flavonoids can promote chemotherapeutic drugs’ sensitivity, indicating a promising potential anti-cancer drug for TNBC.


Since bioavailability is the major challenge for flavonoids, it can be overcome by chemical modification, delivery by nanoparticles, synthetic formulation, and conjugation with metal ions. In conclusion, dietary flavonoids can be exploited for designing therapeutic approaches, in combination or alone, in the prevention and treatment of the clinically challenging cancer, TNBC, through the regulation of miRNAs.


## Figures and Tables

**Figure 1 nutrients-13-01212-f001:**
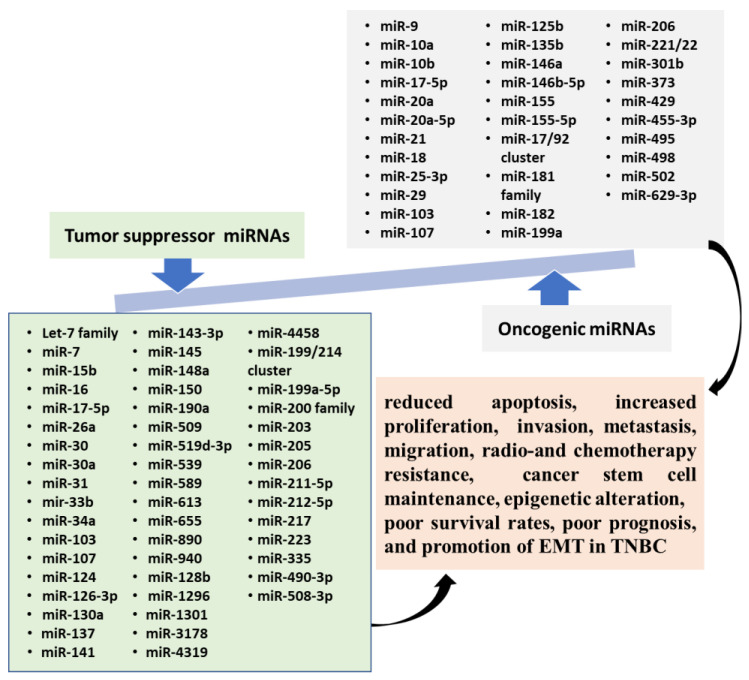
Upregulation and downregulation of specific miRNAs could result in TNBC cell differentiation, cell proliferation, induction of malignant effects, and reduction of apoptosis. Upregulated oncogenic miRNAs enhance EMT and metastasis, induce proliferation, invasion, migration, increase cancer cells’ stemness, and reduce apoptosis [68,69,70,71,72,73,74,75,76,77,78,79,80,81,82,83,84,85,86,87,88,89,90,91,92,93,94,95,96,97,98,99,100,101,102,103]. Oncosuppressor miRNAs were downregulated and are associated with chemoresistance, tumor proliferation, poor survival rates, distal metastasis, poor prognosis, promoting EMT, invasion, migration, and reducing apoptosis. On the contrary, overexpression of these miRNAs represses EMT, suppresses cell growth, proliferation, migration, invasion, tumor cell motility, and promotes apoptosis in TNBC [92,104,105,106,107,108,109,110,111,112,113,114,115,116,117,118,119,120,121,122,123,124,125,126,127,128,129,130,131,132,133,134,135,136,137,138,139,140,141,142,143,144,145,146,147,148,149,150,151,152]. TNBC—Triple negative breast cancer, EM—epithelial-to-mesenchymal transition.

**Figure 2 nutrients-13-01212-f002:**
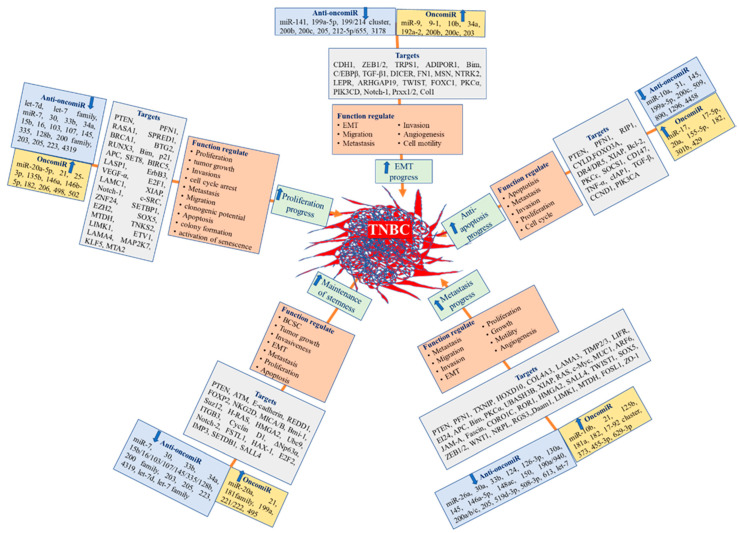
Alteration of miRNAs expression is associated with stemness, EMT, proliferation, metastasis, and apoptosis through targeting several endogenous molecules that contribute to the carcinogenesis of TNBC. PTEN—phosphate and tensin homology, TRPS1—trichorhinophalangeal 1, ADIPOR1—adiponectin receptor 1, CCAAT—enhancer binding protein beta (C/EBPβ), FN1—fibronectin 1, MSN—moesin, NTRK2 or TrkB—neurotrophic tyrosine receptor kinase type 2, LEPR—leptin receptor, ARHGAP19—Rho GTPase activating protein 19, BCSC—breast cancer stem cell, ATM—Ataxia telangiectasia mutated, FOXP2—factor Forkhead-box P2, HMGA2—high-mobility group A protein 2, BT-ICs—breast tumor-initiating cells, Ubc9—ubiquitin-conjugating enzyme 9, ITGB3 (integrin beta3, CSC—cancer stem cell, IGFII—insulin-like growth factor II (mRNA binding protein (IMP3), SETDB1—SET domain bifurcated 1, PFN1—profilin 1, LIFR—leukemia inhibitory factor receptor, EI24—etoposide induced 2.4, APC—adenomatous polyposis coli, XIAP—X linked inhibitor of apoptosis, UBASH3B—ubiquitin-associated and SH3 domain-containing B, v-myc—myelocytomatosis viral oncogene homolog (c-Myc), ZEB1—zinc finger E-box binding homeobox 1, ARF—ARF proteins, MUC1—Mucin1, CORO1C—Coronin 1C, RoR—regulator of reprogramming, ROR1—RTK-like orphan receptor 1, RGS3—G protein signaling 3, BTG2—B-cell translocation gene 2, RUNX3—runt-related transcription factor 3, BIRC5—baculoviral IAP repeat-containing protein 5, LASP1—Lim and SH3 domain protein 1, EZH2—enhancer of Zeste homolog 2, MTDH—Metadherin, AEG-1—astrocyte elevated gene 1, TNKS2—tankyrase 2, LIMK1—LIM kinase-1, LIMK2—LIM kinase-2, ETV1- ETS variant 1, LAMA4—laminin subunit alpha 4, KLF5—Krüppel-like factor 5, MTA2—metastasis-associated protein 2, RIP1—receptor-interacting protein 1, CYLD—cylindromatosis, DR4/5death receptor 4/5, SOCS1—cytokine signaling 1, cIAP1—cellular inhibitor of apoptosis.

**Figure 3 nutrients-13-01212-f003:**
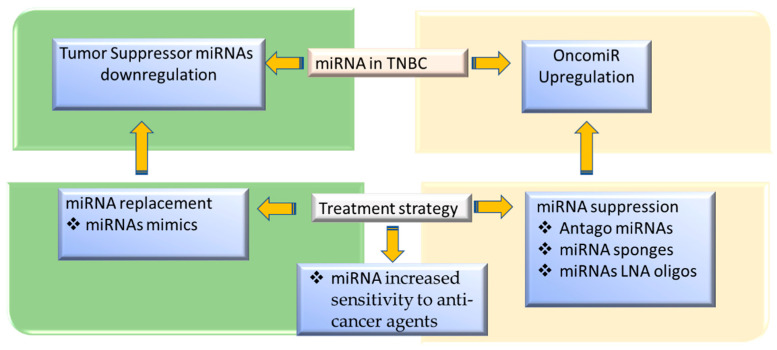
TNBC miRNA therapeutic strategy. Inhibiting the function of OncomiR by the use of miRNAs suppression and promoting the activity of tumor suppressor miRNAs through miRNA mimics can serve as novel therapeutic options against TNBC.

**Figure 4 nutrients-13-01212-f004:**
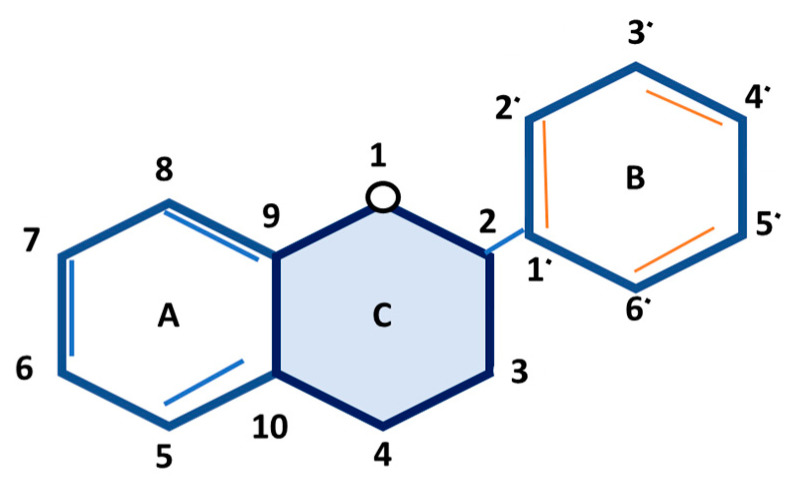
Basic chemical structure of flavonoid. Three rings are normally considered and labeled with letters A, B, and C. The oxygen atom is numbered as the first position in the heterocyclic ring labeled as C. The remaining carbon atoms are numbered from C2 to C10. Ring B shows six positions from C1′ to C6′.

**Figure 5 nutrients-13-01212-f005:**
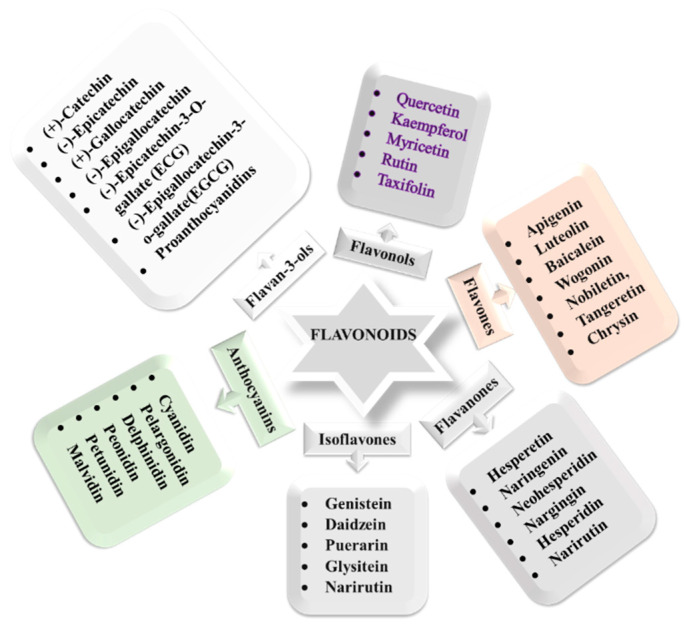
Classification of flavonoids. Based on the degree of oxidation of the central pyran ring flavan, flavonoids are classified into six groups: flavonols, flavanones, isoflavones, flavan-3-ols, flavones, and anthocyanidins. Some common examples are listed under each group.

**Figure 6 nutrients-13-01212-f006:**
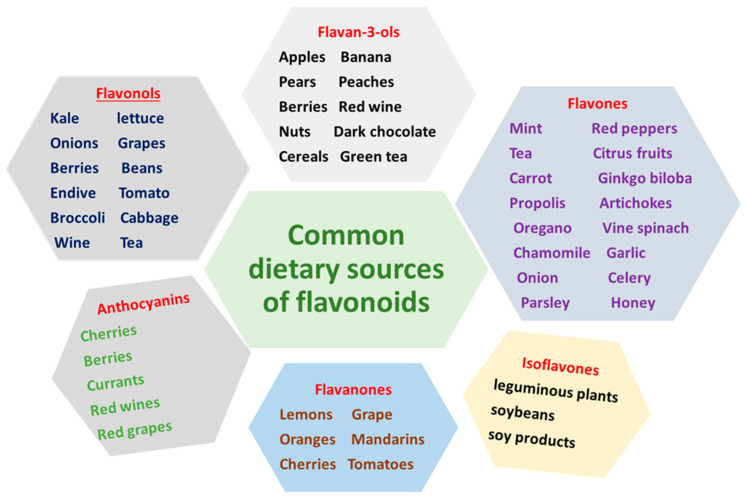
Common dietary sources of flavonoids. Flavonoids are ubiquitously found in edible plants, fruits, and vegetables.

**Figure 7 nutrients-13-01212-f007:**
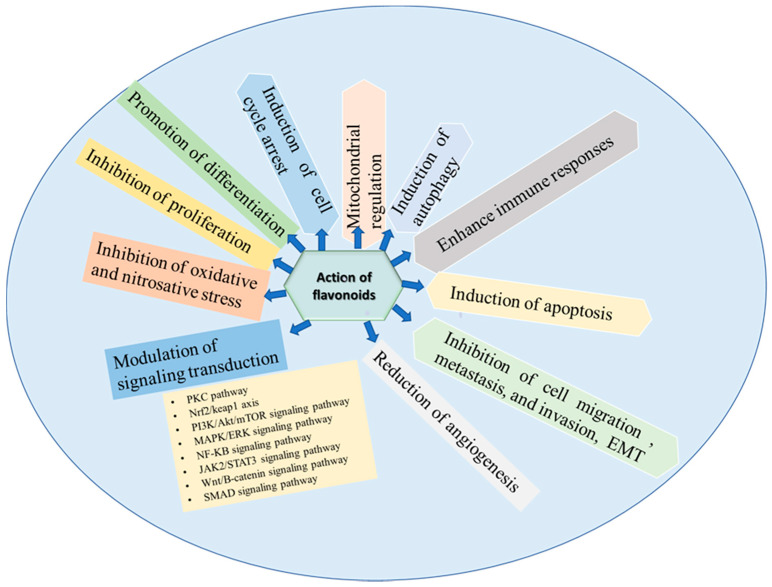
Proposed mechanism of action of flavonoids in the prevention and therapy of TNBC. The anticancer activity of flavonoids is associated with their modulation of signal transduction pathways and inhibition of angiogenesis, proliferation, and metastasis, while promoting apoptosis.

**Table 1 nutrients-13-01212-t001:** Modulation of the signaling transduction pathway by flavonoids in TNBC through targeting micro RNAs. This is the summary table in which the detailed description and references are described in the text.

Flavonoids	Targeted Pathways	Micro RNA Involved in the Pathway	Main Effects of Flavonoids on the Pathway
Quercetin	PKC	miR-200b	Inhibit
Chrysin	Nrf2/keap1	miR-29b-1-5p	Inhibit
Luteolin			Inhibit
Apigenin			Inhibit
Epgallocatechin-3-gallate			Inhibit/Activate
Thymoquinone			Activate
Quercetin	PI3K/Akt	miR-184	Inhibit
Myricetin	MAPK/ERK	miR-489	Inhibit
Fisetin	NF-Kβ	miR-21	Inhibit
Luteolin	JAK/STAT3	miR-18a, mir-30, miR-155, miR-221	Inhibit
Quercetin			
Quercetin	Wnt/β-catenin	miR-1260b	Inhibit
Apigenin			
Fisetin			
Curcumin	SMAD	miR-27a, miR-3613-3P	Inhibit

Abbreviations: PKC—Protein kinase C, Nrf2—Nuclear factor erythroid 2–related factor 2, Keap1—Kelch–like ECH–associated protein 1, PI3K/Akt—phosphatidylinositol 3-kinase/Protein kinase B, JAK/STAT3—Janus-activated kinase pathway/Signal transducer and activator of transcription 3, NF-Kβ—nuclear factor kappa-light-chain-enhancer of activated B cells, Wnt—Wingless and Int–1, MAPK/ERK—mitogen-activated protein kinase/extracellular signal-regulated kinases, SMAD—Smooth-muscle-actin and MAD-related proteins.

## Data Availability

Not applicable.

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
