# Peer review of "The Anticancer Effects of Flavonoids through miRNAs Modulations in Triple-Negative Breast Cancer"

_nutrients, 2021, doi:10.3390/nu13041212_

Round 1

Reviewer 1 Report

The manuscript by Adinew et al describes recent advances in microRNAs and the potential role of flavonoid treatment in cancers and signaling pathways. The manuscript review is compressive and collectively brings together numerous pathways in their relation to various microRNAs. The figures are fairly simple but effective and add to the overall contributions of nutritional supplementation and add benefits of various flavonoids. Overall there is support for the manuscripts with moderate edits in syntax. The manuscript would also benefit from a section describing the effects of flavonoids on cellular/ immune senescence and senescent cells during cancer progression.    

Author Response

Thank you for your comments

Reviewer 2 Report

The authors provide a comprehensive review on current understandings on TNBC and the anti-cancer effect of flavonoids through multiple signaling pathways that involve miRNAs. The article is nicely structured with a few issues. 

1) Proof reading is needed to correct grammar issues throughout the manuscript. For example,” in addition to their anticancer effects by arresting the cell cycle, induce apoptosis, and suppress cancer cell proliferation, the ..” (line 19-20), should be ”…inducing apoptosis and suppressing cancer cell proliferation…” (line 110) “associated” (line 144”) “take place” (line 346)”naturally derived products”. Also, redundancies in the writing should be carefully edited and revised, especially some long sentences with repeating words. Some editing services might be helpful. 

3) Figure descriptions are needed. section 8.2 is too short as a stand-alone section, if it is only used for a summary of an illustrative graph. 

4) section 5.1, 5.2, 9. Long list of miRNAs are presented in the main text. It would be helpful to pick out some examples from the cited literature and emphasize the MOAs. It is not as useful and informational if there is only a list of RNAs. 

Author Response

Dear editor:

We are pleased to resubmit the revised version of the Manuscript ID:  nutrients-1154596. Entitled: " The anticancer effects of flavonoids through miRNAs modulations in triple-negative breast cancer." We appreciate the constructive criticisms, questions, and comments of the reviewers, and we have addressed each of their concerns as outlined below.

Reviewer 2:

  • Proofreading is needed to correct grammar issues throughout the manuscript. For example," in addition to their anticancer effects by arresting the cell cycle, induce apoptosis, and suppress cancer cell proliferation, the." (line 19-20), should be"…inducing apoptosis and suppressing cancer cell proliferation…" (line 110) "associated" (line 144") "take place" (line 346)" naturally derived products." Also, the writing redundancies should be carefully edited and revised, especially some long sentences with repeating words. Some editing services might be helpful. 

     Response: We appreciate and accept the reviewer's suggestion. As advised, the manuscript was revised and edited. Specifically, we made the recommended changes as follows

"…. Flavonoids have shown many biological activities, including antioxidant, anti-inflammatory, and anticancer. In addition to their anticancer effects by arresting the cell cycle, inducing apoptosis, and suppressing cancer cell proliferation, flavonoids can modulate non-coding microRNAs (miRNAs) function" (line 19-20) ". "Several studies report dysregulation of miRNA in tumor-associated macrophages including…" (Line 110). "…. and polyadenylated are take place on pri-miRNA to form precursor microRNA (pre-miRNAs" (line 144). "Naturally derived products play a significant role …" (line 346).

2) Figure descriptions are needed. Section 8.2 is too short as a stand-alone section if it is only used to summarize an illustrative graph. 

Response: As advised, figure descriptions were included. We described figure 1 as "Upregulation and downregulation of specific miRNAs could result in TNBC cell differentiation, cell proliferation, induction of malignant effects, and reduction of apoptosis. Upregulated of oncogenic miRNAs enhance EMT and metastasis, induce proliferation, invasion, and migration, increase cancer cells' stemness, and reduce apoptosis. Oncosuppressor miRNAs were downregulated and are associated with chemoresistance, tumor proliferation, poor survival rates, distal metastasis, poor prognosis, promoting EMT, invasion, migration, and reducing apoptosis. On the contrary, overexpression of these miRNAs represses EMT, suppresses cell growth, proliferation, migration, invasion, tumor cell motility, and promotes apoptosis in TNBC(line 187-193). Figure 3 as "TNBC miRNA therapeutic strategy. Inhibiting the function of OncomiR by using miRNAs suppression and promoting the activity of tumor suppressor miRNAs through miRNA mimics can serve as novel therapeutic options against TNBC" (line 338-340). Figure 4 is described as "Basic chemical structure of flavonoid. The three rings are labeled with the letters A, B, and C. The oxygen atom is numbered as the first position in the heterocyclic ring labeled as C. The remaining carbon atoms are numbered from C2 to C10. Ring B shows six positions from C1` to C6`" (line 358-361). "Figure 5 is described as "Classification of Flavonoids. Based on the degree of oxidation of the central pyran ring flavan, flavonoids are classified into six groups: flavonols, flavanones, isoflavones, flavan-3-ols, flavones anthocyanidins. Some common examples are listed under each group." (line 364-366). Figure 6 is a Common dietary source of flavonoids. Flavonoids are ubiquitously found in edible plants, fruits, and vegetables(line 373-374).  Figure 7 as "Proposed mechanism of action of flavonoids in the prevention and therapy of TNBC. The anticancer activity of flavonoids is associated with their modulation of signal transduction pathways, inhibit angiogenesis, proliferation, and metastasis while promoting apoptosis." (line 823-825 ). We have extended section 8.2 to stand-alone as a section. "Flavonoids are the most popular and the largest plant polyphenols found from the everyday plant-sourced diet. Fruits, vegetables, grains, flowers, stems, roots, bark, and wine are rich natural sources of flavonoids. Studies conducted in 738 men with dietary history to measure the flavonoids quercetin, kaempferol, myricetin, apigenin, and luteolin in foods have shown that tea was the major source that constitutes 61% and vegetables and fruits were about 38% of flavonoids." (line 368-374 )

3) section 5.1, 5.2, 9. The long list of miRNAs is presented in the main text. It would be helpful to pick out some examples from the cited literature and emphasize the MOAs. It is not as useful and informational if there is only a list of RNAs

Response: section 5.1, 5.2 9 has been revised based on the comment as follows and was included in the main text. Section 5.1 "MiR-155 is involved in tumor cell survival, growth, invasion, migration, EMT and immune response. In BC, overexpression of miR-155 significantly inhibited the tumor suppressor gene suppressor of cytokine signaling 1(socs1) and leads to constitutive activation of signal transducer and activator of transcription 3(STAT3) via the Janus-activated kinase pathway(JAK)[58]. In TNBC, miR-155 is overexpressed and involved in metastasis and poor prognosis by inhibiting von Hippel-Lindau (VHL) tumor suppressor expression and induction of angiogenesis. Moreover, miR-155 induced stemness and decitabine resistance by inhibiting the direct target gene tetraspanin-5(TSPAN5) in TNBC. This was further confirmed by the overexpression of TSPAN5 that abolished the effect of miR-155 in enhancing stemness and decitabine resistance in TNBC cell lines. Section 5.2."… Furthermore, overexpression of miR-205 can inhibit cell invasion, differentiation, proliferation through targeting homemobox D10 (HOXD10), which enables to induce the level of P53 and suppress the level of Snail 1 in TNBC cell line." Section 9 "Quercetin strongly repressed tumor metastasis and invasion mediated miRNAs such as miR-146a/b; reduced cell proliferation mediated miRNAs including Let-7 family, as well as induced apoptosis mediated miRNAs such as miR-605. Quercetin also upregulated tumor suppressor miRNAs such as miR-381. Exposure of TNBC cells to resveratrol significantly upregulated tumor suppressor miRNAs such as miR-26a in MDA-MB-231 cells." Please see section 5.1(line 165-173), 5.2(line 180-182) and 9 (line 409-415).

Reviewer 3 Report

The authors have extensively reviewed the role of miRNA in the progression of TNBC. They have also highlighted the potential of flavanoids as a therapeutic against TNBC via regulation of miRNAs. The literature search is through and current. The authors have described the key biological and molecular pathways through which miRNAs can affect tumor suppression or act as oncogene. While the authors have discussed the varied family of flavanoids and showcased some key findings involving these flavanoids in the regulation of miRNAs, the review is lacking how each family of flavanoids can play a role in the regulation of specific pathways through mirNA. The reviewer of the opinion that the readers should have a clear idea about which family of flavanoids can exert which specific pathways to ascertain the true goal of this review paper. An image or a table with the idea suggested above with the information discussed in the review can address this issue.

Author Response

Dear editor:

We are pleased to resubmit the revised version of the Manuscript ID:  nutrients-1154596. Entitled: " The anticancer effects of flavonoids through miRNAs modulations in triple-negative breast cancer." We appreciate the constructive criticisms, questions, and comments of the reviewers, and we have addressed each of their concerns as outlined below.

Reviewer 3

 the review lacks how each family of flavonoids can play a role in regulating specific pathways through miRNA. The reviewer believes that the readers should have a clear idea about which family of flavonoids can exert which specific pathways to ascertain the true goal of this review paper. An image or a table with the idea suggested above with the review's information can address this issue.

Response: We agreed, and we have included one table that addresses the issue. The table is included in the main text. The table's description is as follows "The modulation of signaling transduction pathways by flavonoids is summarized in table 1". Please see line 816-820